# Impact and Implementation of an Early Years Fundamental Motor Skills Intervention for Children 4–5 Years

**DOI:** 10.3390/children11040416

**Published:** 2024-04-01

**Authors:** Clare M. P. Roscoe, Nicola Taylor, Natalie Weir, Robert John. Flynn, Andy Pringle

**Affiliations:** Clinical Exercise Rehabilitation Research Centre, School of Sport and Exercise Science, University of Derby, Derby DE22 1GB, UK; c.roscoe@derby.ac.uk (C.M.P.R.); n.taylor3@derby.ac.uk (N.T.); n.weir2@unimail.derby.ac.uk (N.W.); r.flynn@derby.ac.uk (R.J.F.)

**Keywords:** FMS, physical activity, children, intervention, implementation, evaluation

## Abstract

Fundamental motor skills (FMS) are the cornerstone of a child’s motor development, but concerns remain on the current level of FMS competencies, and intervention is required. This evaluation investigated if a targeted Early Years FMS intervention, delivered by a specialist physical education (PE) provider, improved the FMS of 4–5-year-old children across multiple sites. Methods: The Early Years FMS intervention ran for 18 weeks, 1 h/week, using a standardised programme of activities to develop FMS competencies across 219 children from 15 schools in the Midlands, UK. An adapted assessment was employed as a measure of FMS, assessing locomotor, object control, and stability skills at weeks 1, 9, and 18. The FMS were each rated as green = competent, amber = working towards, or red = not meeting the standards of the skill. A description of key programme implementation characteristics was described. Findings: Statistically significant increases in FMS competencies were achieved for 80% of participants at 18 weeks. Key implementation characteristics for the intervention included consistent staffing, a standardised programme, and a variety of pedagogical approaches delivered by specialist PE staff. Conclusion: This evaluation provided important insights into the effectiveness and implementation of the Early Years FMS intervention to improve FMS competencies in children aged 4–5 years.

## 1. Introduction

Fundamental motor skills (FMS) are the cornerstone of a child’s motor development, encompassing locomotor skills such as running and jumping, object manipulations including throwing and catching, and stability skills such as bending and stretching [1]. As elemental building blocks, FMS serve as precursors to the development of more advanced movement patterns that are essential for sports and physical activity (PA) participation and daily living [2]. Beyond their immediate physical application, these abilities contribute towards the evolution of a child’s physical literacy (PL) and the competence, confidence, and motivation for purposeful PA engagement [3], laying vital foundations that sustain positive PA and health-related behaviours throughout life [4,5]. Crucially, healthy PA in childhood is known to enhance cognition, improve physical, mental, and cardiometabolic health, and prevent harmful weight gain [6]. While the rudiments of motor skills naturally unfold during infancy and early childhood, their complete realisation is dependent on instruction and environmental influences [7,8], as well as participation in a wide variety of rewarding and meaningful movement opportunities [9]. Consequently, it is imperative that FMS is correctly supported in the United Kingdom (UK) to ensure PA and good health and wellbeing in youth and beyond.

Early proficiency in FMS increases the chances of sustained PA and lifelong health [10]. Children should be fully competent in FMS by 7 years of age [11], while UK children aged 5 years and under are recommended to complete 180 min of PA per day, and children aged 5 years and over should aim to achieve 60 min of moderate-to-vigorous PA per day [12]. Yet, a substantial gap exists in UK preschoolers’ FMS ability [13,14] and as few as 10% of preschool children in England meet the Chief Medical Officer’s guidelines for PA [15]. This negative trajectory appears to track into later childhood, and there is perpetual concern that poor motor development and inactivity are exposing children to wide-reaching health consequences [16]. This is highlighted by a recent study in which none of the 219 British primary school participants demonstrated full FMS competence [17]. Moreover, 55% of over 5’s in the UK are classified as inactive [18], contributing significantly to the global obesity crisis that is expected to impact 91 million children worldwide by the year 2025 [19]. Consequently, the life expectancy of the current generation of children is predicted to fall for the first time in modern history [20]. Given these distressing statistics, further research and interventions are imperative to better understand and address FMS development in UK children.

A multitude of innovative research methods that target children’s FMS exists within the literature, including active video games [21,22] and virtual reality [23]. However, although these approaches may build physical fitness, their influence on motor competence remains inconclusive [23,24]. Comparatively, contemporary activities such as cycling and swimming are assumed to enhance coordinated movement patterns [4,25]. However, these ideas have been inadequately explored and are regularly subjected to unreliable assessment methods since they do not readily fit into traditional classifications of FMS [26]. A more accepted mode of FMS intervention that has been thoroughly investigated is unstructured and self-directed free play [27,28]. Nevertheless, despite qualitative data implying gains in physical, cognitive, and emotional development, quantitative data has often been collected via weak-quality studies or has shown non-intervention effects on FMS [29,30]. In contrast, it is argued that significant improvements in motor skills are only achieved through instruction [31]. This notion is supported by a meta-analysis that confirmed that children under instructed conditions improved far beyond those participating in free play [32]. Therefore, structured teaching and guidance may be an important aspect of children’s FMS development.

Early childhood educational environments are considered important spaces for PA attainment and FMS instruction [10] and are also popular interventional settings to increase FMS competency [33,34,35]. In the UK, approximately 100% of 4–5-year-old children are currently in formal childcare or school-based education [36]. Thus, educational settings have access to almost all children including those who are at risk of developmental delay, leaving them uniquely positioned to positively influence FMS ability [37]. Early years physical education (PE) in England is guided by the Early Years Foundation Stage (EYFS) framework [38] and can nurture FMS progression through the provision of appropriate space, equipment, teaching personnel, and opportunities for practice and reinforcement [39]. However, PA and FMS guidance is often entrusted to educators who lack the expertise or confidence to deliver adequate supervision or accurately assess FMS [40]. This issue is further exacerbated by poor school leadership and support [10], larger class sizes and reduced teacher numbers over the last decade [41], insufficient teacher training in FMS, and intense pressure to deliver core academia over PA promotion [40]. In view of the current poor state of children’s FMS in the UK [17] and recent reports of a decline in PA participation around the age of school entry [42], there is a compelling case for providing additional support to teaching staff to enhance FMS outcomes in UK children.

A potentially advantageous method of teacher support may be through the tutelage of skilled external providers [43]. Significant improvements in children’s motor competence have been exhibited both domestically and internationally when teachers have received external training [43] and when teacher delivery has been assisted by trained experts [44]. Equally, there is growing rationalisation within the wider literature that skilled external physical educators, sports coaches, exercise instructors, and motor development experts can directly implement interventions effectively to improve the FMS of early years children [45] Examples of such can be observed in similar studies originating from Europe [46,47], North America [48,49], Canada, [50], and China [51]. Conversely, one UK-based study failed to recreate this success despite having utilised trained external providers [13]. The authors subsequently proposed that this may be because young children could require more targeted skill development activities [13]. Indeed, comparable research has emphasised the importance of manipulating the task and environment to account for individual inabilities and constraints [31]. This has been substantiated by a recent meta-analysis, which concluded that significant enhancements in children’s FMS could be achieved through targeted and developmentally appropriate skill practice overseen by PE specialists within educational settings [52]. Considering this, the FMS of early-year UK children may be best served by the adoption of a similar interventional strategy. The aim of the present study is to investigate if a targeted Early Years FMS intervention delivered by a specialist PE provider can improve the FMS of 4–5-year-old school children.

## 2. Materials and Methods

### 2.1. Participants

Following institutional ethics approval from the University of Derby (45-1819-CRs) and informed consent, children from 15 schools in North Warwickshire and Leicestershire, England, participated in this study. Parental consent was obtained from parents for their children to participate in the Early Years FMS intervention. Furthermore, children’s assent was gained through their desire to be involved in the testing, and those unwilling to participate were removed from the study. The participants were a convenience sample and included 213 school children, aged 4–5 years. The children were from schools in mixed socio-economic status areas, ranging from the 3rd (most deprived) to 8th (least deprived) percentiles on a 1–9 scale for deprivation, according to the index of multiple deprivation [53].

### 2.2. Intervention Context

Coach Unlimited is a PE provider that utilises PE specialists and sports coaches to deliver PE to primary schools across the Midlands, England. In this study, they were responsible for delivering the Early Years FMS intervention, the assessment of FMS at the start of the intervention, and the assessments at weeks 9 and 18 (the final week). The Early Years FMS intervention is a motor skills development programme for Early Years Foundation Stage/reception children in England who are 4–5 years of age. The programme consists of three parts: 1. A screening session, to assist in the early identification of children in need of physical intervention through a series of locomotor, object control, and basic stability movements referred to as FMS/gross motor skills (usually completed early in the school academic year). The FMS were each rated using a traffic light system: green = competent, amber = working towards, or red = not meeting the standards of the skill. 2. Weekly sessions for the identified children use activities based on four different activity types: chase and dodge, ball skills, balance, coordination, and movement. These activities were differentiated according to the ability of the child, allowing the children to develop key physical skills at their own pace. 3. Assessments after weeks 9 (midway) and 18 (end of the intervention) [54].

The selected children with a high red or red/amber profile were identified by the PE provider and reception teachers to follow the 18-week intervention. Initially, 394 children were assessed; of these, 231 children were selected to participate in the Early Years FMS intervention due to their high red or red/amber profile (163 children were eliminated). The final participant sample consisted of 219 children, as some were not present for all assessments. The intervention was run once a week for one-hour by the same three PE provider staff (same member of staff at each school, all with a minimum level 2 National Governing Body qualification, and the company director who oversaw the intervention is a qualified PE teacher). They followed a pre-designed intervention (Appendix A—an example of two sessions) consisting of week one the initial assessment, weeks 2–8 of the intervention, week 9 the mid-way assessment, weeks 10–17 of the intervention, and week 18 the final assessment. Each session had a theme, i.e., Method of Travel, Balance and Linking, Ball Skills, Chase and Dodge, and each specific session followed the same set-up of ‘Starter Activities’ (Musical Statues, Traffic Lights, Row the Boat), ‘Learning Activities’ (Crab Grab, Jumping Jelly, Step On Off and Over, Jump to the Beat, Bouncing Ball, Musical Islands), and ‘Plenary Activities’ (Sleeping Donkey, Stretching, Puppet on a String, Cooldown). Having session plans for each session ensured that all children across the 15 schools were following the same intervention, and this allowed for the analysis of this Early Years FMS intervention to be feasible. Most of the baseline assessments occurred at the start of the child’s academic year (September); this was to try and prevent any improvements in FMS from being attributed to maturation over the academic year. That said, circa 15% of the children (33 children) were assessed in January, which is the start of the second of three main school terms in England. This was due to when the schools opted in and staff availability.

### 2.3. Methodological Context and Assessment of FMS

Children were initially assessed, and they were rated on each of the following FMS: locomotor (running, hopping, forward jump, side stepping, skipping), object control (two-handed catch, underarm throw, overarm throw, kick a ball, alternate dribbling stationary), and stability (one-leg balance, walk along a line, front support, and sideways roll). The children were scored subjectively for each FMS by one of three trained practitioners using a traffic light system: green = competent (consistent success when completing the skill), amber = working towards (inconsistent success within the skill), and red = not meeting the standards of the skill (cannot replicate the skill with any success/likeness). Subjective assessment methods have been adopted by Lindsay et al. (2020) [49] and are considered appropriate for working with early childhood children and, more importantly, their teachers in terms of assessing and understanding the outcomes achieved. The PE provider, in conjunction with the reception class teachers, identified children who were predominantly rated as red for skills and some with a red/amber profile to receive extra support to improve their motor skills development and participate in the 18-week intervention. Approximately one-third of the children from each class participated in the intervention. This is similar to support interventions provided to children across curriculum subjects like English and maths. For context, the red, amber, and green (traffic light system) assessments were utilised because the PE provider aimed for primary school teachers to be in the sessions shadowing them on the first implementation of the 18-week intervention in the school. The aim was that this would allow the teachers to understand the skills being assessed, how to rate the children using a simple traffic light system, and then be able to replicate the intervention for future iterations. It was felt this would allow the Early Years FMS intervention to be sustainable for primary schools and their staff. Unfortunately, due to time restrictions, teachers moving to different schools, and schools opting to use their Physical Education and School Premium to pay for the PE provider to deliver the sessions, this sustainable idea did not transpire.

The Early Years FMS Intervention was designed by the PE provider team of staff, the director of the company has 20+ years of teaching experience, and another member of staff has 10 years of experience in a coaching role and school teaching capacity. These two individuals and other members of the PE provider company were all crucial professional practitioners within the early years and were influential in designing the intervention. The Early Years FMS intervention assessments relate to activities that preschool children participate in and have similarities to the assessments used in the gross motor development 3 (TGMD-3) assessments amongst 3- to 11-year-olds [55]. This study uses the Early Years FMS intervention assessment and intervention as employed by the PE provider and analyses the data they gained.

A partnership evaluation design was adopted, where evaluators work with deliverers to deliver the evaluation. This approach is commonly deployed in assessing the impact and implementation of PA interventions in real-world contexts, including in national and government-funded physical activity programmes [56]. It can be a useful approach when there is no or limited funding to resource a separate ‘consumer evaluation’ that would be undertaken solely by specialist evaluation contractors [56]. Working with deliverers can also help to build evaluation processes into intervention delivery contexts which can help to reduce the burden of an evaluation for participants and deliverers, as well as foster a commitment to evaluate and learn from programme delivery [56]. Institutional ethical approval was obtained for an anonymised data set of measures collected by the delivery staff and transferred securely by the PE provider to the Evaluation Team for analysis. In assuring rigor, a senior member of the research team who had previous experience working with reception-aged children and FMS, as well as delivering the TGMD-2 and TGMD-3 protocols went with the PE provider to a school and observed the assessment of 20 children. The researcher and PE provider practitioner both assessed the children, and the agreement of all children in the 14 skills was 98.4%. This was extremely high, and it was clear to the researchers that the delivery from the PE provider was appropriate and to a very high standard. The children were assessed in small groups (3–4), and the tests were administered on the day by one tester to ensure consistency. A maximum of three coaches performed the assessments in the 15 schools. The assessments took place in the primary school’s hall or an outside area, familiar to the children.

On some occasions, teachers were present with their class for the assessments and programme delivery; however, most of the time, the PE provider and a teaching assistant were the key individuals in attendance. The skills were physically demonstrated and verbally explained to ensure all children had the same clear information on the different skills. If any child did not understand a task correctly, then they were provided with a further verbal description and asked to repeat the trial of the skills [13,15]. The children all had a practice attempt prior to being scored on their assessment; the children were rated in the session as either green, amber, or red. All 14 skills were assessed in a standardised order, and the testing took between 30 and 40 min per group. The skills were always performed in the following order: running, hopping, forward jump, side stepping, skipping, (locomotor skills) two-handed catch, underarm throw, overarm throw, kick a ball, alternate dribbling stationary, (object control skills) one leg balance, walk along a line, front support, and sideways roll (balance).

In assuring rigor, the inter-tester reliability of two schools (18 children) was 90.8% agreement across all 14 skills (range = 88–95%). There was no specified minimum level of percentage agreement; however, 80–85% was deemed acceptable [57] and in line with work more recently by Roscoe et al. (2019) [15]. If at any point the assessors were unsure whether a child had performed a criterion or not, then both the researcher and coach discussed the criteria and agreed on a result. The scores were provided for each skill, per locomotor skills, object control, balance, and total gross motor skill for a child (total FMS).

### 2.4. Statistical Analysis

The results from the traffic light assessment system were converted to a three-point ordinal scale to denote each of the competency levels: 1—not meeting competency; 2—working towards; and 3—competent. Each FMS category is comprised of several subscales (detailed above), which provide total scores and frequency data for each competency level, time point, and FMS category. Descriptive data summarising each FMS category and the total FMS at each assessment were calculated using Microsoft Excel. Friedman’s analysis of variance was carried out using the Statistical Package for Social Sciences (IBM SPSS Version 28, SPSS Inc., Chicago, IL, USA) to compare motor competency scores at three time points: pre-, mid-, and post-intervention, and the alpha level was set a priori at *p* = 0.05. Pairwise comparisons were performed with a Bonferroni correction for multiple comparisons and Kendall’s W to estimate the size of the effect.

## 3. Results

In total, across the 15 schools, 394 children were initially assessed, and 231 participated in the study; however, some were not present for all assessment periods. After excluding these from the study, the final sample consisted of 219 children. Table 1 shows the increase in median scores for each FMS category and total FMS score at each time point during the programme. The median values were derived from the sum of subscales in each FMS category and the total sum of all FMS categories. Results clearly indicate that motor competency scores improved in all three areas of FMS at each stage of assessment.

For each category of fundamental movement skill, the total frequency for each level of criteria achieved at each time point for each subscale were converted to percentages to show the relative number of children achieving each level across the three assessments.

Locomotor skills improved over the course of the programme; Table 2 shows a clear increase in the percentage of children achieving competency, from 42.83% meeting full competency to 88.22% in the final week. Figure 1 illustrates the increase in the number of children achieving full competency from pre-intervention to post-intervention. Friedman’s analysis of variance indicated that locomotion scores were significantly different between time points in the intervention programme χ^2^ (2) = 360.58, *p* < 0.001, W = 0.82. Post hoc analysis revealed statistically significant increases in locomotion scores from pre-intervention (Mdn = 11) to mid-intervention (Mdn = 13) (*p* < 0.001), post-intervention (Mdn = 15) (*p* < 0.001), and from midway (Mdn = 13) to post-intervention (Mdn = 15) (*p* < 0.001).

Object control scores improved over the three time points during the programme; Table 3 shows the relative increase in children achieving full competency from 20.55% at the start of the programme, to 87.21% in the final week. Figure 2 clearly shows the percentage change of children from not meeting competency and working towards it to meeting full competency by the end of the programme. Friedman’s analysis of variance confirmed that object control scores were statistically different between time points in the intervention programme, χ^2^ (2) = 410.01, *p* < 0.001, W = 0.94. Post hoc analysis revealed statistically significant increases in object control scores from pre-intervention (Mdn = 10) to mid-intervention (Mdn = 12) (*p* < 0.001), and post-intervention (Mdn = 15) (*p* < 0.001), and from midway (Mdn = 12) to post-intervention (Mdn = 15) (*p* < 0.001).

Table 4 shows the frequency and percentages of each level of criteria achieved across time points of the programme for stability skills. Skills improved from 35.16% being competent at the beginning of the programme to 96.92% achieving full competency by the end of the programme. Figure 3 presents a clear improvement in stability skills achieved across the programme time points. Friedman’s analysis of variance indicated that stability scores were statistically significantly different between time points in the programme, χ^2^ (2) = 368.51, *p* < 0.001, W = 0.84. Post hoc analysis revealed statistically significant increases in stability scores between all intervention time points, from pre-intervention (Mdn = 9) to mid-intervention (Mdn = 11) (*p* < 0.001) and post-intervention (Mdn = 12) (*p* < 0.001), and between pre-intervention (Mdn = 9) and post-intervention (Mdn = 12) (*p* < 0.001).

Finally, when the total sums of FMS scores were analysed across the three time points, Freidman’s analysis of variance indicated significant differences, χ^2^ (2) = 428.50, *p* < 0.001, W = 0.97. Post hoc analysis revealed statistically significant increases in total FMS scores from pre-intervention (Mdn = 30) to mid-intervention (Mdn = 35) (*p* < 0.001), post-intervention (Mdn = 41) (*p* < 0.001), and midway (Mdn = 35) to post-intervention (Mdn = 41) (*p* < 0.001).

## 4. Discussion

The aim of this study was to investigate if the targeted intervention delivered by specialist PE providers can improve the FMS of 4–5-year-old school children. Key findings emerging from this investigation support the idea that significant increases in FMS proficiency can be achieved through a specific Early Years FMS intervention for most participants at the end of the 18 weeks. This aligns with the findings in the literature [33,34,35]. This study also provides important insights into the design and implementation of the intervention, including the assessment of FMS competencies, standardised clear and logical programme of delivery by the same staff over 18 weeks. Given the absence of studies that report on programme content [35], these insights are important in facilitating effectiveness and future learning.

### 4.1. Effectiveness of the Early Years FMS Intervention

In view of the current poor state of children’s FMS in the UK [17], and recent reports of a decline in PA participation around the age of school entry [42,58], there is a compelling case for providing additional support to teaching staff to enhance FMS outcomes in UK children. Hulteen et al. (2018) [4] found that gross motor skill development and PA were positively associated with a child’s PL. Developing PL in the early years can have positive outcomes for cognitive function, academic achievement, and social-emotional development [59]. Low PA and PL in young children can have negative health consequences, increasing the risk of obesity, cardiovascular disease, and mental health problems [60]. Thus, effective interventions can help young children not only develop the abilities and skills needed to engage in physical activity but also to lead healthy lives in the future.

The results show the increase in median scores for each FMS category and total FMS score at each time point during the intervention. The results indicate that motor competency scores improved in all three areas of FMS at each stage of assessment (Table 1). However, for most children to meet full competency across all three FMS components, engagement in the full 18-week intervention was required, indicating the importance of maintaining the participation of children for the lifespan of the intervention. This aligns with a recent study of a similar design that successfully enhanced all aspects of children’s FMS over a 16-week period [46]. Conversely, positive gains in motor competency have been observed in equivalent studies of considerably shorter study length [47,48,49], although this is possibly due to them having adopted partial test protocols that omitted either object control or locomotor skills and therefore required less time compared to the present study, which addressed FMS practice more holistically. In helping to facilitate, the Early Years FMS intervention was delivered using a variety of activities focusing on different aspects of motor skill development and integrated structured learning with playfulness to create an engaging environment for developing FMS in 4–5-year-olds Notably, comparable interventions have highlighted the value of enjoyment and playful actions during structured FMS practice [46,47,49], and would seem to be an integral component of motor development in young children. Furthermore, this kind of deliberate play (a type of practice that is purposeful and systematic) has been found to increase motor flexibility and intrinsically motivate children to be physically active [61].

Further, regarding the prescriptive element, Van Capelle et al. (2017) [34] completed a meta-analysis of 20 studies to quantify the improvement in FMS after interventions in 3–5-year-old children. It was subsequently concluded that children should practice at least three times per week for 30 min per session. Similar recommendations were proposed by Engel et al. (2018) [33] following a review of FMS interventions for 3–12-year-old children, and this approximately corresponds with several gainful interventions delivered by external providers around the world [47,48,49,50,51]

Yet, in contrast to this, the Early Years of FMS intervention reported improvements in FMS in fewer weekly sessions. However, FMS has previously benefited from a wide range of interventions, irrespective of substantial variations in prescription [62]. Johnson et al. (2023) [63] have also stressed that the quality of implementation may be of more significance than dosage, so it is difficult to draw meaningful conclusions regarding the overall impact of the session frequency. Moreover, in the absence of a comparator group, it is unclear if outcomes were due to the intervention or external influences and activities that may have been undertaken outside of the intervention. Therefore, further investigations are needed to confirm this outcome.

### 4.2. The Implementation and Evaluation of the Early Years FMS Intervention

In facilitating the impact of the Early Years FMS intervention on FMS competencies, the design and the implementation of an intervention is an important consideration [35]. This is particularly important when the literature [64] notes a lack of information on the content of interventions. In the current study, the Early Years FMS intervention was delivered by a specialist PE provider and included a range of activities that provided variation and progressive challenges for participants. Varied and progressively more challenging activities have been similarly adopted in comparable FMS interventions [46,51]. It has been reasoned that in the absence of this approach, children are at risk of stagnating and becoming automated to activity as they gradually adapt to and successfully perform a task [51] so it is vital to increase the complexity of an activity in line with a child’s learning [46].

The sessions were organised logically, with starter, main, and cool-down exercises accompanied by clear session plans which were replicable over the 18 weeks. This type of structure is in line with other FMS interventions [46,50,51] and is important for both children and deliverers when building familiarisation and the opportunity for most children to improve over the intervention period. In facilitating the necessary continued engagement of children in the intervention, staff employed a variety of strategies such as analogues, story narrative, game-based approaches, and use of cues and feedback which offers the opportunity of creating variety and to help make the intervention content both interesting and consumable to children. Moreover, sessions focused on different areas of FMS to help build variety which could help facilitate engagement [65,66]. In the delivery of physical activity programmes, staffing has been identified as an important consideration by the Department of Health [67]. In this study, the intervention was run weekly for an hour per session by the same three specialist staff across the 15 schools. This approach helps to build relationships between children and coaches and allows coaches to become familiar with what children could achieve and were capable of with support. Moreover, the intervention was delivered by a highly trained specialist PE provider who had experience in providing FMS curriculum and had good insight into which pedagogical approaches were effective and how to engage children. They also had insight into what content had been delivered on a weekly basis, and the potential to develop more bespoke strategies to facilitate engagement on an individualised basis. This child-centered approach has been successfully implemented in previous interventions where FMS learning and physical activity are adjusted to meet the needs of the learners [63].

Pringle et al. (2020) [56] identify that it is important to learn from the process of implementing evaluations, including what works well and why, and what works less well, in order that lessons can be learned. Key considerations in this study were the use of a simplistic traffic light system of assessment to benchmark and report participants’ FMS proficiency (Green = competent, Orange = working towards, Red = not meeting). This method arguably serves as a simple, innovative approach to assess FMS in 4–5-year-olds. The approach also enables a targeted and tailored support system for children who may require additional assistance, potentially leading to more significant improvements in FMS. This traffic light system also offers a straightforward and easily understandable visual representation of a child’s FMS proficiency, making it feasible for educators, parents, and other stakeholders to understand and potentially be trained on. While simplicity is an asset, there is a risk of oversimplification. The traffic light system may not capture the nuances and subtleties of a child’s FMS development, potentially overlooking specific strengths or areas for improvement within each category. A simple/clear intervention to follow, clear assessments, and a simple traffic light system rating for teachers/practitioners to adopt are important, especially for practitioners who face time and resource pressures to deliver interventions. Lindsay et al. (2020) [49] similarly administered a subjective assessment in response to a lack of resources that prevented the adoption of commonly used tools such as the TGMD, and this facilitated the involvement of teaching staff in the programme.

As indicated in previous studies, FMS guidance is often entrusted to educators who lack the expertise or confidence to deliver adequate supervision or accurately assess FMS [40]. This issue is further exacerbated by poor support [10], larger class sizes, and reduced teacher numbers over the last decade [41]. Therefore, the need for clear guidance and local resources that are easily deployed in schools is paramount. As such the protocol used in this intervention may also be valuable for non-specialist PE staff. This has been demonstrated in other countries where instructors have trained and partnered with teaching staff to assist with intervention delivery so that implementation may continue after the cessation of the intervention [46,49]. A further important implementation consideration was the assessment of children using small groups involving 3–4 children by a limited number of staff might help to provide a more inclusive experience. However, thinking about potential scaling up, if this form of assessment was more widely used there may be variability in how educators interpret this system–introducing an element of subjectivity. This subjectivity could impact the accuracy and reliability of the assessments. As such any wider implementation of the system needs to be supported with appropriate training, education, and resources for those staff members involved in delivering interventions [10,68].

### 4.3. Limitations and Strengths of This Evaluation

It is important that there is both reflection and learning from conducting evaluations [69]. The limitations of this research are that gender was not reported in this study, and we might expect to see improvements in locomotive skills for girls and object control for boys [14,70,71]. Further, young children 4–5 years old are in a natural period of biological and age-related growth and development [72,73]; therefore, we need to be mindful of how we differentiate between natural progression or consequences of the intervention which contribute to the improvement in children’s FMS. The long-term impact of selective participation on overall FMS development and future PA would benefit from longer-term investigation to determine the effectiveness and sustainability of this approach since there is evidence to suggest that children do not always fully retain gains in FMS post-intervention [48]. Moreover, the validity and sustainability of interventions have been challenged by research due to a paucity of follow-up studies [11,74]. This longer-term investigation should also include a comparator or control arm of the intervention to confirm if any improvements in competencies are due to the intervention and where possible, the use of independent researchers to collect FMS measures where the resources exist. A further acknowledgement is that the traffic light system of assessing the children’s skills was simplistic and not to the standard of assessments such as the TGMD-3, however, as previously stated this manner of scoring the children was originally used with the aim of sustainability and the primary schools/teachers adopting this intervention and assessment in the future, unfortunately, this did not transpire. That said, this process is simple for teachers to understand when the PE provider feeds back to them, and it allows them to classify each child. Furthermore, alignment was shown with some of the assessment measures observed in this current study e.g., running, throwing, and hopping with the measures used by the TGMD-3, highlighting the relevance for the assessment of Early Years children of this Early Years FMS intervention. The limitations are balanced by research strengths, including a standardised programme of delivery for participants and the development of a real-world evaluation design implemented across 15 delivery sites in 18 weeks. A partnership evaluation design was also adopted [56], whereby the PE deliverer worked with independent evaluators to undertake evaluation tasks such as in-building data collection to programme delivery, so these processes become streamlined and the transfer of anonymised data for analysis. This is common in real-world investigations, including those cited in the peer-reviewed literature [56,67,75,76]. Also, the use of independent measures and processes for the measurement of FMS competencies in three categories. In this study, the inclusion of information on the content of the intervention and how this was implemented is especially important given that research [35] has noted the absence of the reporting of interventions in the literature and viewed as ‘good practice’ [56,67,69]. A further strength is the inclusion of insights into how evaluation methods were also put into practice, which serve to assist service providers and evaluators with the planning of future activities. The inclusion of content in implementation and evaluation is important to avoid unnecessary replication and a failure to learn from previous experiences [56,76].

## 5. Conclusions

Significant increases in FMS proficiency can be achieved through the Early Years FMS intervention for most participants at the end of the 18 weeks. This proves that the Eary Years FMS intervention delivered by the PE provider is therefore extremely effective for 4–5-year-old children in the UK. Key implementation characteristics are important considerations due to their impact on intervention effectiveness.

## Figures and Tables

**Figure 1 children-11-00416-f001:**
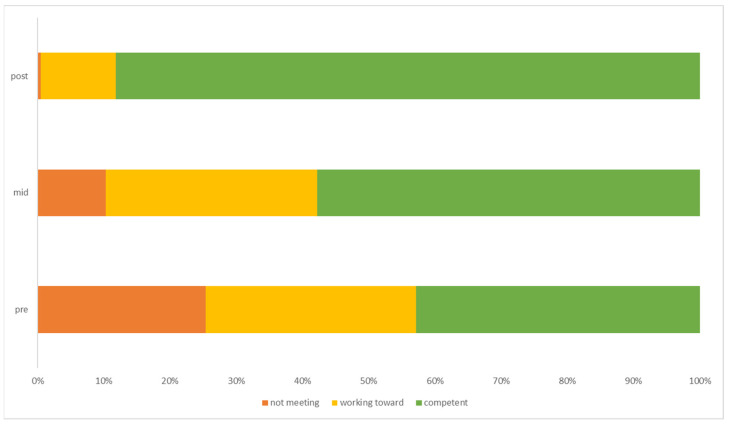
The percentage of children achieving each level of criteria for locomotor skills across each time point of the intervention.

**Figure 2 children-11-00416-f002:**
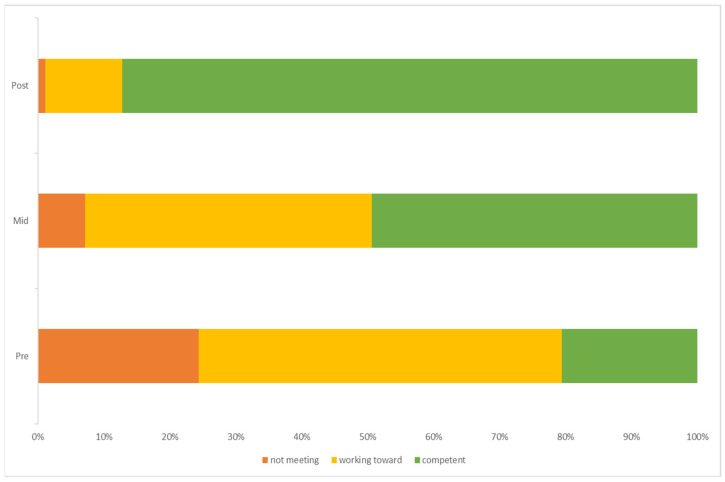
Percentage of children achieving each level of criteria for object control skills across each time point of the intervention.

**Figure 3 children-11-00416-f003:**
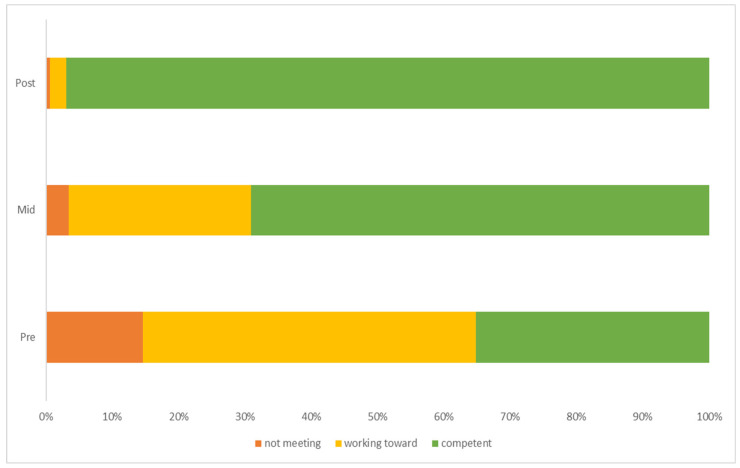
Percentage of children achieving each level of criteria for stability skills across each time point of the intervention.

**Table 1 children-11-00416-t001:** Median scores for each FMS category and total FMS at each intervention time point.

	Pre-Intervention	Mid-Intervention	Post-Intervention
Locomotor	11	13	15
Object Control	10	12	15
Stability	9	11	12
Total FMS	30	35	41

**Table 2 children-11-00416-t002:** Frequency and percentage of children meeting each level of criteria for locomotor skills across each intervention time point.

	Pre-Intervention		Mid-Intervention		Post-Intervention	
	Frequency	%	Frequency	%	Frequency	%
Not meeting	278	25.39	113	10.32	5	0.46
Working towards	348	31.78	349	31.87	124	11.32
Competent	469	42.83	633	57.81	966	88.22

**Table 3 children-11-00416-t003:** Frequency and percentage of children meeting each level of criteria for object control skills across each intervention time point.

	Pre-Intervention		Mid-Intervention		Post-Intervention	
	Frequency	%	Frequency	%	Frequency	%
Not meeting	267	24.38	78	7.12	12	1.10
Working towards	603	55.07	477	43.56	477	11.69
Competent	225	20.55	540	49.32	955	87.21

**Table 4 children-11-00416-t004:** Frequency and percentage of children meeting each level of criteria for stability skills across each intervention time point.

	Pre-Intervention		Mid-Intervention		Post-Intervention	
	Frequency	%	Frequency	%	Frequency	%
Not meeting	128	14.61	30	3.42	5	0.57
Working towards	440	50.23	241	27.52	22	2.51
Competent	308	35.16	605	69.06	849	96.92

## Data Availability

Data is unavailable due to privacy or ethical restrictions.

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
