# Peer review of "Impact and Implementation of an Early Years Fundamental Motor Skills Intervention for Children 4–5 Years"

_children, 2024, doi:10.3390/children11040416_

Round 1

Reviewer 1 Report

Comments and Suggestions for Authors

This paper employs intervention experiments to investigate interventions for children's fundamental movement skills (FMS). The results indicate a significant improvement in FMS abilities for 80% of participants by the 18th week. Key features of the intervention include consistent staffing, standardized protocols, and a variety of teaching methods provided by professional sports staff. I believe several modifications should be made before publication in the following areas:

Regarding the research topic, interventions for children's fundamental movement skills are a hot topic, making this study valuable.

In the introduction and discussion sections, there is insufficient discussion of similar studies, and the advantages of the intervention measures provided by the authors compared to those in existing research should be further elucidated and supplemented.

In terms of research methodology, this paper adopts intervention experiments to design a series of intervention measures to demonstrate the impact of interventions on children's fundamental movement skills. The research methodology employed is suitable and appropriate, ensuring the reliability of the research conclusions. However, the author's description of the sample and research methods is rather brief. Further elaboration is needed on how the sample was selected, when it was selected, how experimental and control groups were established, and how the accuracy of the research results was controlled.

Regarding the research conclusions, this paper provides new insights for developing a series of intervention measures to improve children's fundamental movement skills. The author could strengthen the educational practice aspects of the methods proposed and further elucidate their role in educational practice.

Author Response

Regarding the research topic, interventions for children's fundamental movement skills are a hot topic, making this study valuable

-

Thank you for your positive feedback 

In the introduction and discussion sections, there is insufficient discussion of similar studies, and the advantages of the intervention measures provided by the authors compared to those in existing research should be further elucidated and supplemented.

3

We have added/amended within the last paragraph of the introduction to make clearer what we had already started to introduce and added additional references to support our point:  We have added the following

“Equally, there is growing rationalisation within the wider literature that skilled external physical educators, sports coaches, exercise instructors, and motor development experts can directly implement interventions effectively to improve the FMS of early years children [47]. Examples of such can be observed in similar studies originating from Europe [Battaglia et al., 2019; 46], North America [45, Lindsay et al., 2020], Canada [Roach & Keats, 2018], and China [Zhang et al., 2022].”

The research methodology employed is suitable and appropriate, ensuring the reliability of the research conclusions. However, the author's description of the sample and research methods is rather brief. Further elaboration is needed on how the sample was selected, when it was selected,

-

4

5

Thanks for the confirming suitability of the methods and reliability of the conclusions drawn.

We have already explained in the text/methodology that the sample is a convenience sample, but we have added the following into the ‘Intervention Context’ section in the Methods to make it clearer when the sample was selected: “The majority of the baseline assessments occurred at the start of the child’s academic year (September), this was to try and prevent any improvements in FMS being attributed to maturation over the academic year. That said, circa 15% of the children (33 children) were assessed in January which is the start of the second of three, main school terms in England, this was due to when the schools opted in and staff availability.”

This was also added in the ‘Methodological context and Assessment of FMS’ section as it supports the use of the traffic light system and helps to clarify the reason a simple assessment was utilised: “For context, the red, amber, green (traffic light system) was utilised because the PE provider aimed for primary school teachers to be in the sessions shadowing them on the first implementation of the 18-week intervention in the school. The aim was this would allow the teachers to understand the skills being assessed, how to rate the children using a simple traffic light system and then be able to replicate the intervention for future iterations. It was felt this would allow the Early Years FMS intervention to be sustainable for primary schools and their staff. Unfortunately, due to time restrictions, teachers moving to different schools and schools opting to use their Physical Education and School Premium to pay for the PE provider to deliver the sessions, this sustainable idea never transpired.”

How experimental and control groups were established, and how the accuracy of the research results was controlled.

12

This is an evaluation of an FMS intervention in a real-world setting and so no control and experimental group was established.  This is a good point and so in the strengths and limitations section, we had already a sentence suggesting that future research could employ a control and experimental design.  

Reviewer 2 Report

Comments and Suggestions for Authors

Dear authors,

Congratulations for your work. An impressive support to run this study!

My only concern is about children's consent to participate in this study. They are only 4 - 5 years old. I wonder if their parents were aware of the Smart MOVES Programme.  

Please describe how this relationship was managed, with the children's parents, from an Ethical perspective. 

Good luck with your next research!

Author Response

Congratulations for your work. An impressive support to run this study!

Thank you for your positive feedback!

My only concern is about children's consent to participate in this study. They are only 4 - 5 years old. I wonder if their parents were aware of the Smart MOVES Programme.

3

In Materials and Methods:  We have made some additions to what was already included to answer this valid comment: “Parental consent was obtained from parents for their children to participate in the Early Years FMS intervention. Furthermore, children’s assent was gained through their desire to be involved in the testing and those unwilling to participate were removed from the study.”

Please describe how this relationship was managed, with the children's parents, from an Ethical perspective

3

In Materials and Methods:  As per the above comment we have provided, in the Participants section within the Methods we have included about consent from parents for their children to participate in the Early Years FMS intervention.

Reviewer 3 Report

Comments and Suggestions for Authors

It is an honor to review a good research paper.

The research design was appropriate and the research results were meaningful. However, we are requesting some modifications, so please revise and supplement them.

Previous studies on children in the UK have been analyzed in Introduction. Please analyze and add prior studies on child FMS in various regions, including Europe and Asia, besides the UK. Also, we are only presenting the positive aspects of children's physical activity, but please add information about the outcomes of children's physical activity.

2.2 Is SM Athletic Development Program and Support Program developed by professional researchers? Please add relevant information. Was it made by a simple physical education major? Is it made by a physical education professor? Do researchers make it?

A total of 394 children participated in the results, but only the final 231 were selected for the study, so please add information on why 163 students were eliminated.

365 Line Applicable Employees [10;
61] -> [10,61]

In the discussion, it is being developed appropriately with previous studies supporting the results of this study. However, previous studies have mainly focused on studies of children in the UK, so please add information on how FMS for children is operating in different countries.

Author Response

It is an honour to review a good research paper.

The research design was appropriate, and the research results were meaningful. However, we are requesting some modifications, so please revise and supplement them.

-

Thank you very much for your positive comments, it is very pleasing to learn that you feel the research design is appropriate and the results meaningful.   Thank for your suggestions in refining our manuscript.

Previous studies on children in the UK have been analysed in Introduction. Please analyse and add prior studies on child FMS in various regions, including Europe and Asia, besides the UK.

 Also, we are only presenting the positive aspects of children's physical activity, but please add information about the outcomes of children's physical activity.

3

3

We have added/amended within the last paragraph of the introduction to make clearer what we had already stared to introduce and added additional references to support our point: “Equally, there is growing rationalisation within the wider literature that skilled external physical educators, sports coaches, exercise instructors, and motor development experts can directly implement interventions effectively to improve the FMS of early years children [47]. Examples of such can be observed in similar studies originating from Europe [Battaglia et al., 2019; 46], North America [45, Lindsay et al., 2020], Canada [Roach & Keats, 2018], and China [Zhang et al., 2022].

We have touched on outcomes in the introduction such as “Crucially, healthy PA in childhood is known to enhance cognition, improve physical, mental, and cardiometabolic health, and prevent harmful weight gain” but also added the below to reinforce outcomes as well as reiterate negative consequences as well as positive;”

We have included “Hulteen et al. (2018) found that gross motor skill development and PA were positively associated with a child’s PL.  Developing PL in the early years can have positive outcomes on cognitive function, academic achievement, and social-emotional development (Donnelly & Lambourne, 2011).  Low PA and PL in young children can have negative health consequences, increasing risk of obesity, cardiovascular disease, mental health problems (Janssen et al, 2016) thus effective interventions can help young children not only develop the abilities and skills needed to engage in physical activity and lead healthy lives in the future”. 

SM Athletic Development Program and Support Program developed by professional researchers? Please add relevant information. Was it made by a simple physical education major?

 Is it made by a physical education professor? Do researchers make it?

3

4

Thank you for this comment, we have now added the following description in the ‘Methodological Context and Assessment of FMS’ section: “The Early Years FMS intervention was designed by the PE provider team of staff, the director of the company has 20+ years teaching experience and another member of staff has 10 years of experience in a coaching role and school teaching capacity. These two individuals and other members of the PE provider company are all crucial professional practitioners within the Early Years and were influential in designing the intervention.”

Thank you for your suggestion. Under ‘Intervention Context’ the following has been added to support the qualifications the PE provider staff have: “The intervention was run once a week for one-hour by the same three PE provider staff (same member of staff at each school, all with a minimum level 2 National Governing Body qualification and the Company Director who oversaw the intervention is a qualified PE teacher) and they followed a pre-designed programme (supplementary material 1 – example of two sessions) consisting of…”

A total of 394 children participated in the results, but only the final 231 were selected for the study, so please add information on why 163 students were eliminated.

4

Thank you for this comment, in addition to the explanation around the Red, Ambeer Green [RAG] system profile the following has been added in the ‘Intervention Context’ section within the Method:

“Initially 394 children were assessed, of these, 231 children were selected to participate in the Early Years FMS intervention due to their high red or red/amber profile (163 children were eliminated). The final participant sample consisted of 219 children as some were not present for all assessments.”

65 Line Applicable Employees [10;61] -> [10,61]

We have clarified this point in the text. 

In the discussion, it is being developed appropriately with previous studies supporting the results of this study. However, previous studies have mainly focused on studies of children in the UK, so please add information on how FMS for children is operating in different countries.

10-11

10-11

Thank you for this suggestion we have added additional content on studies to the discussion. We have said that “This child centered approach has been successfully implemented in previous interventions where FMS learning and physical activity is adjusted to meet the needs of the learners (Johnson et al., 2023).”    

Reviewer 4 Report

Comments and Suggestions for Authors

Dear authors,

Introduction

The introduction provides an in-depth justification for the research study using contemporary, relevant studies to support claims. 

The challenges and questions around the involvement of external providers of PE in schools are an important topic of debate. 

Materials and Method

This section of the paper is very concerning for me. I see the potential for the risk of bias and lack of independence in the selection of participants and also when assessing the students, due to the heavy involvement by a company delivering PE services. Furthermore, I feel that throughout this paper the name of the company is used continuously which raises the question, why? The name of the company needs to be removed from the paper. This research has sadly exposed itself to a high level of bias as can be inferred from Line 171 "A partnership evaluation design was adopted, where evaluators work with deliverers to deliver the evaluation [53]." The testing needed to be conducted by independent research assistants. 

My other major concern is that the intervention and the testing lack detail and there is no information provided to suggest that the testing or method of assessment is valid and reliable in this age group or any other age group. 

Based on the serious issues identified, I feel that I might be right in suggesting that this research was just a feasibility study. 

Results

It is difficult to assess the quality of the results because there are little or no details about how the assessment data scores were determined. 

Discussion

Generally well-written, however difficult to assess based on the methodology issues. 

Funding

Your paper states there was no external funding. Was there in-kind support from Coach Unlimited, for which services were provided at no cost to the researchers?

This area of research is important, however there are serious flaws in the research. 

Kindest regards

Author Response

Reviewer 4

Thanks for your comments. Your suggestions along with the other reviewers strengthen our paper

Methods

This section of the paper is very concerning for me. I see the potential for the risk of bias and lack of independence in the selection of participants and when assessing the students, due to the heavy involvement by a company delivering PE services.

Furthermore, I feel that throughout this paper the name of the company is used continuously which raises the question, why? The name of the company needs to be removed from the paper.

This research has sadly exposed itself to a high level of bias as can be inferred from Line 171 "A partnership evaluation design was adopted, where evaluators work with deliverers to deliver the evaluation [53]."

The testing needed to be conducted by independent research assistants.

1

Throughout

12

We have removed the name of the company and the intervention from the paper including the intervention name in the title of the paper.  

Following MDPI guidelines we aimed to be transparent about the name of the company, as if we had omitted their name, we would not have acted in an open manner.  However, I think it is important to mention that we have been transparent about the contribution of Coach Unlimited who delivered the intervention. This is important in providing a context to the intervention that can help inform the reader of how the intervention was implemented.    This is not an uncommon approach in the reporting on the evaluation of physical activity interventions in community contexts, including those that have been published in the peer-review literature and we cite examples of the literature already in the paper e.g. 67, 76.

Thanks for this, we have removed the name of the deliverer.

Regarding why, it is not uncommon for deliverers to work in collaboration with evaluators, including interventions that are delivered across multi-sites and in some cases, national physical activity programmes as well as those that have been published in the international peer-review literature.   We refer to some of these studies and sources that demonstrate case studies of partnership evaluation designs that are already in the manuscript 56 67, 75-76.   

While the testing by independent researchers is ‘an ideal’, it is not always possible to service such ‘consumer’ evaluation designs [56], including in the evaluations of interventions in real world contexts. However, we have added your specific suggestion of deploying independent researchers to collect data where this is possible. This is now located in the strengths and limitations which is in the discussion, thanks for this suggestion.

My other major concern is that the intervention and the testing lack detail and there is no information provided to suggest that the testing or method of assessment is valid and reliable in this age group or any other age group.

4

Appendix

Page 4

12

We have supported our work with the following passage in the ‘Methodological Context and Assessment of FMS’ section: Subjective assessments methods have been adopted by Lindsay et al. (2020) [49] and considered appropriate for working with Early Years children and more importantly their teachers in terms of assessing and understanding the outcomes achieved.

Intervention details are provided as exemplar session plans in the appendix, description in the ‘Intervention Context’ section.   

I have added this in the ‘Intervention Context’ section in the methods: “For context, the red, amber, green (traffic light system) was utilised because the PE provider aimed for primary school teachers to be in the sessions shadowing them on the first implementation of the 18-week intervention in the school. The aim was this would allow the teachers to understand the skills being assessed, how to rate the children using a simple traffic light system and then be able to replicate the intervention for future iterations. It was felt this would allow the Early Years FMS intervention to be sustainable for primary schools and their staff. Unfortunately, due to time restrictions, teachers moving to different schools and schools opting to use their Physical Education and School Premium to pay for the PE provider to deliver the sessions, this sustainable idea never transpired.”

We have further added to the limitations section where we acknowledged the following: “A further acknowledgement is that the traffic light system of assessing the children's skills was simplistic and not to the standard of assessments such as the TGMD-3, however, as previously stated this manner of scoring the children was originally used with the aim of sustainability and the primary schools/teachers adopting this intervention and assessment in the future, unfortunately this did not transpire. That said, this process is simple for teachers to understand when the PE provider feeds back to them, and it allows them to classify each child. Furthermore, alignment was shown with some of the assessment measures observed in this current study e.g. running, throwing, hopping with the measures used by the TGMD-3, highlighting the relevance for assessment of Early Years children of this Early Years FMS intervention.”

It is difficult to assess the quality of the results because there is little or no details about how the assessment data scores were determined.

6

I have added the following to the data analysis section page 6:

Each FMS category comprised of several subscales (detailed above) which provided total scores and frequency data for each competency level, time point and category.  

And the following to Results:

The median values were derived from the sum of subscales in each FMS category and the total sum of all FMS.

Added ‘for each subscale’ under table 1 for clarity.

Generally well-written,

however difficult to assess based on the methodology issues.

-

Thank you for your positive comments about the writing style, this is consistent with the views of the three independent reviewers who also peer-reviewed our manuscript.

Your paper states there was no external funding. Was there in-kind support from Coach Unlimited, for which services were provided at no cost to the researchers?

.

13

There was no external funding to pay for the time involved in data preparation, analysis, reporting, writing and review, nor the cost of access to software, library resources and writing up which has all been provided internally through our university. 

In the funding declaration we have added in that data was collected as part of programme delivery.  In line with our institutional ethical approval, the PE Provider transferred the anonymized data set to the evaluators for analysis.   We already indicate that this task was performed in the manuscript.  

This area of research is important, however there are serious flaws in the research

5

We are pleased to see that you think this research is important, thank you very much. 

As we have already indicated in the paper, we explain why it is important to evaluate community-based physical activity programmes.   We already explain in the paper that investigations of this nature provide important insights around impact and implementation of interventions in real world contexts in order that learning can take place which informs and shapes practice.

Round 2

Reviewer 1 Report

Comments and Suggestions for Authors

Thank you to the author for the very detailed revision. The author has responded to the previous questions, and I believe it is now acceptable for publication.